# TOWARDS BY-DESIGN INTERPRETABLE TRANSFORMERS VIA MODULAR INTERPRETABILITY-GUIDED ALIGNMENT

## ABSTRACT

Transformer models offer strong predictive performance but generally lack interpretability, limiting their adoption in high-stakes applications such as neuroscience. Existing explainable AI methods tend to produce inconsistent and biologically ungrounded explanations, which reduce their usefulness in uncovering condition-specific mechanisms. In this paper, we propose an Interpretability-Guided Alignment module designed to enhance the explainability of pre-trained Transformer models by aligning their internal representations, weights with established external biological knowledge. We introduce a novel conditional interpretable layer and a block-wise interpretability mechanism that provide localized, human-understandable insights into model decisions. Experimental evaluation on two real-world Alzheimer's disease datasets, namely Seattle and ROSMAP, demonstrates that our approach not only achieves strong classification accuracy but also uncovers biologically meaningful interpretations by identifying key pathways supported by external biological databases such as KEGG and WikiPathways, thereby outperforming existing baselines. Specifically, our solution achieves more than three times higher biological interpretability scores for the Alzheimer's disease condition compared to existing methods. Furthermore, our approach has the potential to enhance the interpretability of other Transformer-based models across application domains when integrated with relevant external knowledge.

## 1 INTRODUCTION

Large Language Model (LLM) architectures have gained substantial traction due to their ability to efficiently solve a wide range of tasks (Touvron et al., 2023a; Achiam et al., 2023; Jiang et al., 2024). Despite their considerable success in prediction tasks, Transformer-based models are inherently difficult to interpret. Although attention mechanisms form the foundation of these architectures, relying on attention weights for interpretability is limited and sometimes misleading (Jain & Wallace, 2019). Moreover, interpreting the role of MLP components remains a challenge.

Explainable AI (XAI) comprises complementary techniques that enhance transparency and trustworthiness, helping researchers better understand how models reason about predictions (Nasarian et al., 2024; Tjoa & Guan, 2020; Mersha et al., 2024). A recent discussion (Hendrycks & Hiscott, 2025) highlighted the challenges of mechanistic interpretability (Lieberum et al., 2023; Bolukbasi et al., 2021). Earlier approaches, including feature visualizations (Mordvintsev et al., 2015) and saliency maps (Adebayo et al., 2018), yielded inconsistent explanations. As an alternative, a top-down interpretability approach that focuses on internal representations has recently been advocated by Hendrycks & Hiscott (2025).

In neuroscience, the combined capabilities of Transformers and XAI techniques can be leveraged to effectively analyze biological datasets and clarify the reasoning behind model outputs, adding an important layer of support for trust and clinical relevance (Thirunavukarasu et al., 2023; Heimberg et al., 2025). Some researchers have explored attention visualization and post-hoc XAI techniques, such as Integrated Gradients (IG) (Theodoris et al., 2023; Chen et al., 2023; Cui et al., 2024; Yang et al., 2022; Heimberg et al., 2025). However, these explanations lack consistency and do

not reflect established biological knowledge, limiting their usefulness in uncovering disease-specific mechanisms. These limitations highlight the need for by-design interpretable methods that produce meaningful explanations without relying on additional XAI techniques.

In particular, attention mechanisms (Yang et al., 2022; Cui et al., 2024; Theodoris et al., 2023; Ye et al., 2025; Zhang et al., 2024; Vaswani, 2017) and XAI methods (Sundararajan et al., 2017) fail to provide biologically grounded explanations. To address this, we propose an effective solution to make pre-trained Transformers interpretable through the integration of external biological knowledge. Our proposed module analyzes the Transformer block outputs, capturing rich, context-dependent token representations. It leverages a conditional interpretable layer dynamically activated by disease context, to produce biologically meaningful token importance scores aligned with established neuroscience facts using context-specific disease knowledge. Our main contributions are as follows:

- We propose an Interpretability-Guided Alignment module, designed to be integrated into a pre-trained Transformer architecture to render it interpretable by design.
- We design a novel interpretable conditional layer that makes the internal representations, block outputs and weights interpretable in a disease-specific context.
- We devise a block-wise interpretability mechanism that converts contextualized token representations from individual blocks into human-understandable interpretations, unlocking block-specific insights.
- We validate the effectiveness of our approach using two real-world single-nucleus RNA sequencing (snRNA-seq) datasets focused on Alzheimer's disease. The results provide evidence that it is able to uncover biologically meaningful interpretations that standard Transformer architectures and XAI methods fail to capture.

## 2 RELATED WORK

**Transformer-based architectures** for single-cell data are designed to analyze the activity levels of thousands of genes measured in single cells. These models are based on the attention mechanism, and they operate at different token levels: some treat individual genes as tokens, others model biological pathways, and a few represent entire cells as sentences. At the gene level, models encode cells as sequences of genes, producing embeddings for downstream tasks. Examples include Geneformer (Theodoris et al., 2023), scBERT (Yang et al., 2022), scGPT (Cui et al., 2024), CellFM (Zeng et al., 2025), Scale-free and Unbiased Transformer (Zhang et al., 2025), and scFoundation (Hao et al., 2024). At the pathway level, TOSICA (Chen et al., 2023) represents each cell as a sequence of biological pathways. Other approaches focus on alternative formulations. For instance, CellPLM (Wen et al., 2024) treats cells as sets of tokens and tissues as sentences while capturing cell–cell relationships. Cell2Sentence (Levine et al., 2024) converts gene expression profiles into cell sentences by ranking genes according to expression levels. While these models demonstrate strong predictive performance, their interpretability remains limited. Attention scores have been employed to highlight biologically relevant features, but the explanations often lack consistency and fail to integrate established biological knowledge. This motivates the need for by-design interpretable approaches.

**Explainable AI** encompasses several techniques devised to help in understanding the reasoning behind models' predictions. Many popular techniques have been explored, including gradient-based attribution approaches such as Integrated Gradients (IG) (Sundararajan et al., 2017) and Input × Gradient (Ancona et al., 2017), relevance-propagation methods such as DeepLIFT (Shrikumar et al., 2017) and Layer-wise Relevance Propagation (LRP) (Bach et al., 2015), and game-theoretic approaches such as Shapley Additive Explanations (SHAP) (Lundberg & Lee, 2017). Variants such as DeepLIFT SHAP and Gradient SHAP extend these principles by approximating Shapley values using gradient or DeepLIFT-based attributions.

In the literature, scSimilarity (Heimberg et al., 2025) model for cell analysis employs an MLP backbone with a decoder and uses the IG (Sundararajan et al., 2017) technique for explanation. TOSICA (Chen et al., 2023) relies on attention scores (standard self-attention mechanism) to interpret biological knowledge at the pathway level, while scBERT (Yang et al., 2022) and scGPT (Cui et al., 2024) adopt attention scores to interpret at the gene level. In this paper, we critically evaluate popular XAI techniques, and attention mechanisms (Ye et al., 2025; Zhang et al., 2024; Vaswani, 2017) adopted

in prior studies. Despite their prevalence, these methods often yield inconsistent and biologically uninformative interpretations, showing weak alignment with ground-truth biological knowledge, echoing recent critiques on the reliability of post hoc explanations (Hendrycks & Hiscott, 2025). To address these limitations, we propose a framework that renders transformers interpretable, producing biologically valid and context-aware insights while maintaining strong predictive performance.

## 3 PROPOSED METHOD

### 3.1 MODEL OVERVIEW

We build on a Transformer backbone composed of multiple standard blocks. Each block includes an attention mechanism, a feed-forward network, residual connections, and layer normalization. Our goal is to render this architecture interpretable by ensuring that each block's output contributes meaningfully to the model's explanations aligned with external knowledge (e.g., biology). To this end, we propose an Interpretability Module with a novel conditional shared layer that quantifies contextualized embeddings, activations, and weights to estimate token importance.

Our framework follows four steps. First, we fine-tune a Transformer backbone for the downstream task. Next, we integrate our proposed interpretability module into the Transformer block outputs. We then apply Interpretability-Guided Alignment learning to jointly optimize the predictive performance and explanation quality. Finally, we generate grounded, domain-specific interpretations by analyzing internal representations, intermediate block outputs, and weights. This design ensures that interpretations are grounded in the model's learned representations and constrained by external biological knowledge, making them both relevant to the model's decision process and valuable for scientific insights without relying on post-hoc explanation techniques.

In this paper, our module operates on contextualized token embeddings, where the input tokens correspond to biological pathways denoted $P$. Each pathway vector is computed by applying a binary mask to the cell's gene expression vector, retaining the expression values of genes associated with the pathway while setting all other genes to zero. This masked vector serves as the input token for the pathway. In our setup, each cell contains 5029 highly variable genes, with 216 pathway tokens from WikiPathways database and 142 from KEGG database.

### 3.2 PROPOSED INTERPRETABILITY MODULE: ARCHITECTURE AND COMPONENTS

We integrate the proposed Interpretability module into a trained Transformer $Model^{Trained}$ consisting of $N$ blocks. The module is organized into three sub-modules: block-level, model-level, and output-level. Each Transformer block is connected to its corresponding block-level sub-module, which projects the block's contextualized token embeddings into interpretable scalar values. The outputs from all block-level sub-modules are then aggregated and adjusted by the model-level sub-module, and the output-level sub-module produces the prediction with its associated interpretation. Figure 1 details the Transformer and interpretability module components.

1. **Block-level sub-module:** Each Transformer block outputs a sequence of contextualized token embeddings. The block-level sub-module converts each vector embedding into a single scalar value per token, producing a vector of dimension equal to the number of input tokens. Formally, given a sequence of $(|P| + 1)$ token embeddings $E_{block_j}$ from the $j$-th block, the sub-module transforms it into an interpretable $(|P| + 1)$-dimensional vector $X_{block_j} \in \mathbb{R}^{(|P|+1)}$, where each element corresponds to an input token. This process is defined as follows:

$$X_{block_j} = submodule\_block_j(E_{block_j}) \tag{1}$$

Here, the sub-module $submodule\_block_j$ consists of a shared projection layer (i.e., 1D convolution layer) followed by our proposed Shared Conditional interpretable layer described below. Each Transformer block is connected to its corresponding block-level sub-module. By generalizing across all $N$ blocks, the outputs of the $N$ sub-modules can be written as follows:

$$X_{block_j} = (X_{block_j}, j = 1, \ldots, N) \tag{2}$$

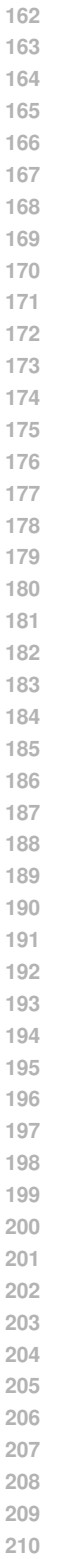
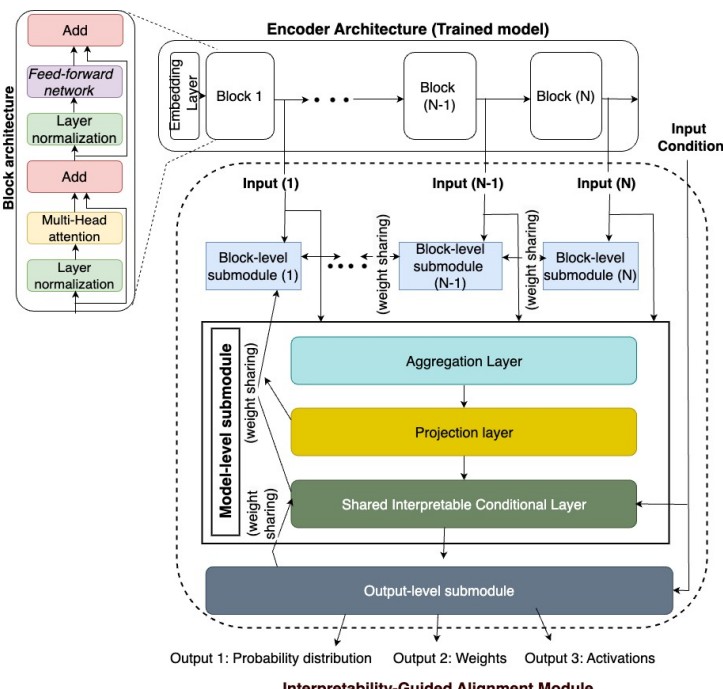

Figure 1: Interpretable Transformer architecture based on Interpretability-Guided Alignment Module

The outputs from each block-level sub-module reveal what the block has captured with respect to the external knowledge, providing interpretable, token-level insights across the entire model.

2. **Model-level sub-module:** It takes $N$ inputs derived from $N$ Transformer's blocks and combines them using a token-wise aggregation strategy. The aggregated representation is then fed into a shared projection layer followed by our proposed shared conditional layer, along with an input condition. This layer produces a semantic vector that aggregates information from all blocks while preserving token-level interpretability. Each element of the vector represent a token and its biological importance to the model's prediction. Higher values reflect greater relevance. The intuition behind this module is that each block captures aspects of token relationships, from low-level to high-level. By aggregating these representations, the model-level sub-module summarizes all learned information per token, ensuring that the contributions from all blocks are reflected in the final interpretation.

- **Aggregation Layer**: This layer takes as input the outputs of $N$ Transformer blocks, denoted as $(E_{block_j}, j = 1, \ldots, N)$, and performs token-wise aggregation by summing the corresponding token representations across blocks. This operation is written as follows:

$$Agg_{internalrep} = Aggregation\_layer(E_{block_j}, j = 1, \ldots, N) = \sum_{j=1}^{N} E_{block_j} \quad (3)$$

where $Agg_{internalrep}$ is the resulting aggregation of the different blocks' semantic representation vectors.

- **Projection layer**: This is a 1D convolution layer, shared across all the $N$ block-level sub-modules and Model-level sub-module. It transforms the $(|P| + 1)$ internal vector embeddings into $(|P|+1)$-dimensional vector. This process can be defined as follows:

$$Proj = Projection\_layer(Agg_{internalrep}) \quad (4)$$

Here, each element of $Proj \in \mathbb{R}^{(|P|+1)}$ corresponds to a specific token.

- **Shared Conditional interpretable layer**: This proposed layer receives two inputs. The first is a condition variable referred to as $C_i$, a binary indicator variable estimated based on the context of the i-th observation. The second input is its associated projected internal vector $Proj_i$ or the projected tokens extracted from the different $N$ block-level sub-modules at the inference time. To handle data depending on the context, this layer is extended by two weight matrices, denoted $W_{global} \in \mathbb{R}^{(|P|+1) \times (|P|+1)}$ and $W_{context} \in \mathbb{R}^{(|P|+1) \times (|P|+1)}$. The first one, is a global matrix applied to any observation regardless of its context. In contrast, the second one is designed to be activated or deactivated dynamically based on the context of input tokens, as determined by the condition $C_i$. It is followed by SeLU activation, resulting in a vector of size $(|\mathcal{P}| + 1)$. The weights and output activations are context-aware. This process is defined as follows:

$$CI_{proj_i} = Shared\_layer(X_{block_j}, C_i)$$
$$= SeLU\left(Proj_i W_{global} + C_i \cdot (Proj_i W_{context})\right) \quad (5)$$

Here $Proj_i W_{global}$ is equivalent to a transformation based on a conventional fully connected layer, while the second term $C_i \cdot (Proj_i W_{context})$ is activated only when the context of input tokens is deemed important. Otherwise, it remains deactivated. The term $CI_{proj_i} \in \mathbb{R}^{(|P|+1)}$ is the semantic internal representation.

3. **Output-level sub-module:** This layer carries out three different operations to generate three distinct outputs. It adopts three components: a softmax layer that predicts the class probability, a shared conditional layer that provides the second output, and an aggregation that summarizes the estimated weight matrices extracted from the shared conditional layer. This process can be expressed as follows:

$$Output_1 = Prob_i = Softmax(Dense(CI_{proj_i}))$$
$$Output_2 = Activations_i = CI_{proj_i} \quad (6)$$
$$Output_3 = Weights_i = Aggregation(W_{global} + C_i W_{context})$$

Where $Output_1 = Prob_i \in \mathbb{R}^k$ refers to the inferred probability distribution over the $k$ classes. $Dense(.)$ is a fully connected layer. $Output_2 \in \mathbb{R}^{(|P|+1)}$ denotes the second output presenting the interpretable semantic representation. $Output_3 \in \mathbb{R}^{(|P|+1)}$ refers to the aggregation of a set of weight matrices, where the contribution of each matrix is dynamically modulated based on its activation under the input condition $C_i$ which determines whether a weight matrix is activated or deactivated. $W_{global} \in \mathbb{R}^{(|P|+1) \times (|P|+1)}$ and $W_{context} \in \mathbb{R}^{(|P|+1) \times (|P|+1)}$ are weights matrices.

## 3.3 INTERPRETABILITY-GUIDED ALIGNMENT WITH EXTERNAL KNOWLEDGE: LEARNING FOR JOINT PREDICTION AND EXPLANATION

Let $GT_{label}$ and $GT_{tokens}$ denote the ground truth one-hot encoded label vector and the ground truth token annotations vector, respectively. Specifically, $GT_{labels}$ represents the true class label for a cell, while each element of the vector $GT_{tokens}$ corresponds to a discrete annotation for a specific input token. This step focuses on tweaking the parameters of proposed interpretability module while leaving all others unchanged. Besides, we add a second loss function denoted $Loss_{interpretability}$ that aligns the model with external knowledge. It is expressed as follows:

$$Loss_{interpretability} = Loss_{weights} + Loss_{activations}$$
$$Loss_{activations} = MSE(Output_2, GT_{tokens}) \quad (7)$$
$$Loss_{weights} = MSE(Output_3, GT_{tokens})$$

Essentially, at this step, the model is trained to optimize both the cross-entropy loss, denoted $Loss_{cassification}$ and defined by $CE(.)$, and the interpretability loss, $Loss_{interpretability}$, which is based on the mean squared error $MSE(.)$. The total loss function is written as follows:

$$Loss_{global} = Loss_{cassification} + Loss_{interpretability}$$
$$Loss_{cassification} = CE(Output_1, GT_{label}) \quad (8)$$

Here $CE(.)$ is the cross-entropy function. Note that the interpretability-guided alignment step is performed using a randomly sampled subset comprising 50% of the total training observations.

### 3.4 Token-Level Interpretations: Generation from Model and Blocks Internal Representations

After the Interpretability-Guided Alignment with External Knowledge step, the interpretations of each Transformer block output as well as a global interpretation are derived from the activations and the weight matrices extracted from the interpretability module. Formally, the interpretations for the $i$-th input token sequence at the model-level and the block-level are as follows:

- **Model-level interpretations** based on activations ($Interp_{activations}$) and weights aggregation ($Interp_{weights}$):

$$Interp_{weights} = rank(Weights_i) = rank(Aggregation(W_{global} + C_i W_{context})) \quad (9)$$

$$Interp_{activations} = rank(Activations_i) = rank(CI_{proj_i}) \quad (10)$$

- **Block-level interpretations** derived from the internal representations of each Transformer block:

$$(Interp_{activations^{block_j}}, j = 1, \ldots, N)$$
$$Interp_{activations^{block_j}} = rank(submodule_{block_j}(E_{block_j}))) \quad (11)$$

Here, $rank(.)$ is a function used to quantify the importance of tokens sorted in descending order, high value indicates more important token. $C_i$ denotes the prediction produced by the vanilla trained Transformer model without using our interpretable module. $Interp_{activations^{block_j}}$ refers to the partial interpretation that quantifies the importance assigned to different tokens by the j-th transformer block.

## 4 Experiments

In this section, we first introduce the two real-world datasets. We then describe the methodology and evaluation metrics, followed by the state-of-the-art techniques used for comparison. Next, we present the experimental results. The hyper-parameter settings, and additional implementation details are presented in the appendix.

### 4.1 Datasets

The experiments are performed using two benchmark single-cell datasets, namely Seattle and ROSMAP. Below are detailed descriptions of each dataset:

1. **Seattle dataset** (Gabitto et al., 2024): This is a benchmark dataset consisting of millions of labeled cells. We used 84,164 single-cell transcriptomes, each cell is labeled as either Alzheimer's Disease (AD) or Control (normal). The distribution of labels is balanced.

2. **ROSMAP dataset** (Mathys et al., 2019): This refers to the so-called, Religious Orders Study or the Rush Memory and Aging Project (ROSMAP), a real-world single-cell RNA sequencing dataset made up of cells sourced from human brain donors. Each cell is classified into one of two categories: Control or AD. It contains 61,990 cells with a balanced label distribution.

### 4.2 Methodology and evaluation metrics

In this paper, each dataset is randomly divided into 2 partitions, training, and test sets. The test set is employed for evaluating the model's performance, it comprises 20% of cells. The training set consists of 80% of cells. This process is repeated five times, then the average is computed.

We assess both predictive and interpretability performance. The predictive performance is evaluated using the classification accuracy, while the biological interpretability is assessed using a pathway-focused protocol. The model ranks input tokens by importance scores, and we measure how well the top-ranked tokens align with ground-truth AD-related pathways established through GWAS genes, reflecting well-established findings in neuroscience. For an AD cell, a higher value indicates that biologically meaningful tokens are correctly identified. For a control cell, a lower value reflects better performance, as AD-related pathways are not detected. Further evaluation and implementation details are provided in the Appendix.

### 4.3 BASELINES

We evaluated the performance of our proposed approach against attention mechanisms and explainable methods used in previous works (Heimberg et al., 2025; Yang et al., 2022; Chen et al., 2023).

1. **LLM-originated attention mechanisms:** We adopted a transformer architecture and employed various attention mechanisms proven to enhance LLMs performance, including Differential Attention (Ye et al., 2025), Attention Steering (Zhang et al., 2024), and standard Multi-Head Self-Attention (MHA) (Chen et al., 2023; Vaswani, 2017).

2. **Explainable attribution-based methods:** We compared our approach with several gradient-based XAI methods, including Integrated Gradients (Sundararajan et al., 2017), DeepLIFT (Shrikumar et al., 2017), GradientSHAP (Scott et al., 2017), and Input×Gradient (Simonyan et al., 2013). We also included random guessing as a baseline, which randomly assigns attribution scores to the set of tokens.

3. **Baseline layers:** We evaluated the interpretability of activations of our proposed shared conditional interpretable layer against two popular alternatives Mixture-of-Experts (MoE) (Fedus et al., 2022) and the standard fully connected (FCL) layer. This later is by default used with standard multi-head self-attention.

4. **Our proposal:** $Interp_{activations}$ and $Interp_{weights}$ denote the interpretations based our proposed interpretable transformer derived from the activations and weights, respectively.

### 4.4 EXPERIMENTAL RESULTS

In this paper, given the availability of external knowledge for Alzheimer, our ultimate goal is to enhance the interpretability of transformers in the context of Alzheimer's disease. We consider improvements in interpretability valuable as long as the new predictive performance remains indistinguishable from the baseline. There is currently no pre-trained LLM for Alzheimer's disease (AD) classification that process sequences of tokens (pathways), based on the same set of biological pathway databases (KEGG 2021 and WikiPathways 2024 Human) explored in this study. We defined our own transformer architecture inspired by (Chen et al., 2023; Touvron et al., 2023b;c) trained from scratch. The experimental results are organized as follows:

**Predictive performance analysis:** Figure 2 presents the comparative results in terms of classification accuracy on the Seattle and ROSMAP data sets. It depicts the performance of the model using different types of attention mechanisms within each transformer block, specifically standard self-attention (MHA), differential attention, and attention steering. It also compares the Mixture-of-Experts against our proposed conditional interpretable layer. The model-based standard self-attention and our proposal provide nearly the same performance in terms of accuracy, which proves using the proposed interpretability module does not decrease the performance.

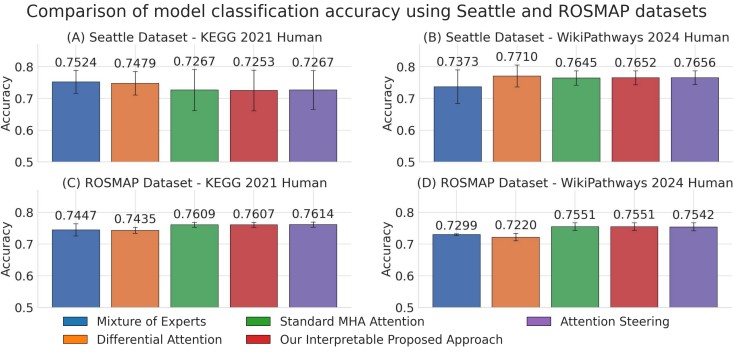

Figure 2: Predictive performance analysis

**Evaluation of activations interpretability:** Figure 3 compares the interpretability of internal representations extracted from the different baselines, where the set of input tokens is represented as an encoded vector, the i-th element represents the i-th input token (biological pathway). We compute the intersection of the most important tokens (highest scores) with ground-truth token annotations.

In general, for AD predicted cells, a higher number of detected tokens (pathways proven biologically to be related to Alzheimer) indicates better performance, while for control cells, a smaller value is better and outlines that AD-related pathways are not selected as the most important tokens, which is aligned with biological external knowledge. It is clear that our proposal consistently outperforms all the baseline approaches in interpretability, providing evidence that the model produces biologically meaningful activations.

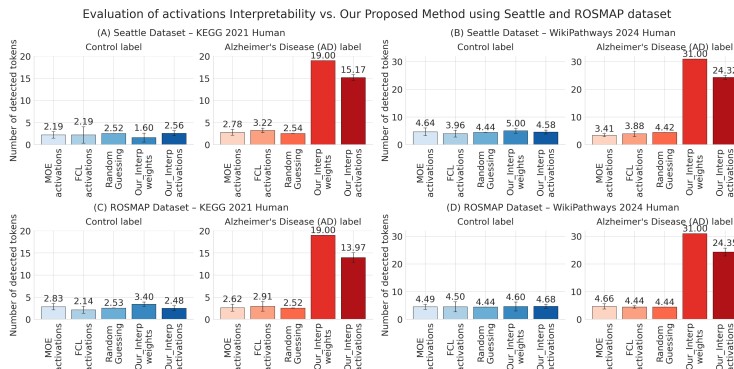

Figure 3: Evaluation of internal representations interpretability

**Evaluation of interpretations from LLM-Originated Attention Mechanisms:** Figure 4 compares a set of attention mechanisms designed for improving LLMs, which are shown to be better than standard multi-head self-attention. So, we extracted the attention matrices that represent the interactions between the input tokens. Then, we aggregated them appropriately across heads and tokens to get a vector of size equal to the number of input tokens. In general, an important token has high attention score values while an irrelevant one has lower values. Then, we quantified their relevance biologically. From the results, our proposal and the attention steering were able to pinpoint a high number of pathways for the AD condition on various datasets. However, for the control condition, the attention steering is the worst among all the baselines due to the bias toward AD, i.e., the same tokens are pinpointed for both conditions. While our proposal based on weights or activations pinpointed a lower number of pathways, which indicates its ability to dynamically adapt its weights and internal representations according to the context of input and prediction thanks to the proposed conditional interpretable layer.

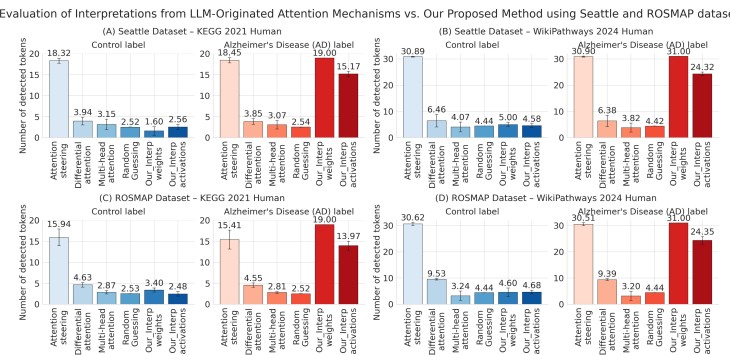

Figure 4: Evaluation of Interpretations from LLM-Originated Attention Mechanisms

**Evaluation of XAI attribution-based methods explanations:** Figure 5 shows the comparison of explanations generated using a set of post-hoc XAI gradient-based methods vs. our proposal. The experimental results provide clear evidence that our proposal surpasses all the other competitors in interpretability. It identified the most important tokens depending on the predictions, which is not the case with XAI methods even though they outperform random guessing.

**Evaluation of block-wise vs model-wise sub-modules' internal representations interpretability:** Figure 6 displays the comparison of interpretations extracted from each transformer block and quan-

tified using our proposed module, denoted block-wise. The model-wise interpretation is extracted directly from the interpretability module. The model-wise sub-module yields more interpretable and informative activations, outperforming block-wise activations in capturing biologically meaningful attributions across both datasets. But, when it comes to each block individually, we notice the same pattern on the ROSMAP dataset using KEGG with WikiPathways, and Seattle with KEGG, for AD condition, block 1 pinpoints a higher number of meaningful tokens than block 2, while for control condition, the block 2 is the best, this may be considered as during the learning process each block has a specialized-role depending on a specific condition whether AD or control. While, the results based on Seattle with WikiPathways was an exception, where block 1 was better at identifying pathways related to both conditions, followed by block 2 that yielded very close results.

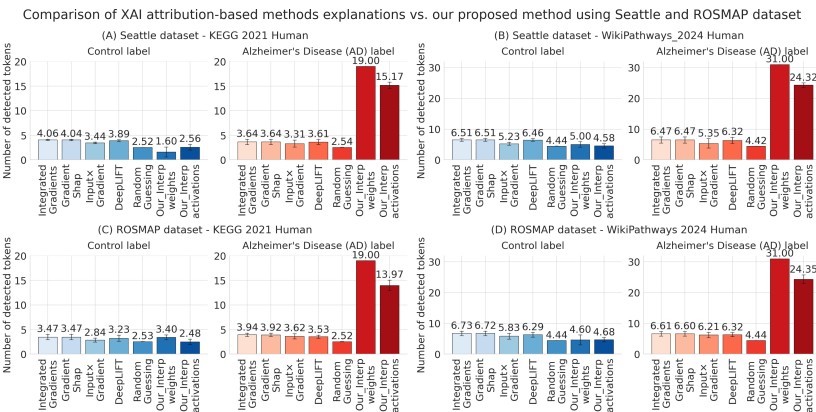

Figure 5: Evaluation of XAI attribution-based methods explanations:

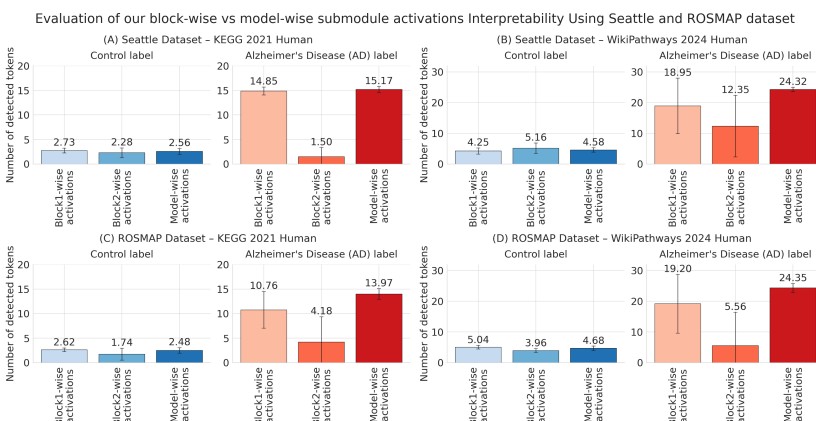

Figure 6: Evaluation of block-wise vs. model-wise sub-modules internal representations interpretability

## 5 CONCLUSION

In this work, we introduce an Interpretable Transformer based on Modular Interpretability-Guided Alignment, designed to classify input tokens and provide predictions with biological interpretations. By structuring input cells using external biological knowledge, we represent tokens based on biological pathways. The Transformer's interpretability is achieved by aligning token importance with relevant biological pathways. Experiments on two benchmark datasets confirm that our approach produces context-aware, biologically valid interpretations while maintaining predictive performance, addressing a key limitation of most existing XAI methods. In future work, we plan to extend our framework to LLMs to explore their scalability and adaptability in more complex architectures.

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

## A  APPENDIX

This supplementary material provides additional details regarding the evaluation metrics, hyper-parameters settings, and implementation details.

### A.1  EVALUATION METRICS

In this work, we assessed the performance of models using the following:

**Classification accuracy:** It is used to compare the discriminative capacity of Transformer models. A high accuracy value means better performance.

**Interpretability task:** we defined our own protocol aligned with biology, to evaluate the learning ability of the model at different levels block and model as well, quantify the importance of tokens, and generate context-aware interpretations with respect to the predicted conditions. Let $k$ denote the number of relevant tokens, biological AD-related pathways that contain at least one gene identified in Genome-Wide Association Studies (GWAS). For the i-th input cell, $Inter_i$ refers either to an interpretation deduced from activations (internal representations), or Explanations generated using a post-hoc XAI method, or attention scores matrices, aggregated appropriately, where each element represents a score assigned to a specific token.

Formally, let $G_{\text{tokens}}$ be the set of $k$ ground-truth pathways associated with the Alzheimer's disease (AD) condition. We quantify the quality of the interpretations by computing the number of correctly identified pathways as follows:

$$R = \frac{\sum_{i=0}^{M} |G_{tokens} \cap \text{top-}k(Inter_i)|}{M} \tag{12}$$

Here, for the AD condition, $M$ is the number of cells predicted as AD. A higher $R$ value indicates that more important tokens are (AD-related pathways) are correctly detected. Meanwhile, for the control condition, $M$ is the number of cells predicted as control, and a smaller $R$ value reflects better performance, as it indicates that pathways related to the AD condition were not detected.

### A.2  HYPER-PARAMETERS SETTINGS

In this paper, several experiments were conducted to select the optimal hyper-parameters. The total number of highly variable genes is 5029 per cell. The embedding size for tokens equals 128. For the Mixture of experts, shared conditional layer, or dense layer, the number of neurons equals the number of input tokens. The number of epochs for pre-training the model was set to 3. The batch size is set to 256. The number of heads is set to 2. For the parameter $N$, we tested various number of blocks between 2 and 10, based on the results using our datasets, it turns out small number of block leads to better classification accuracy. The learning rate is fixed at $1 \times 10^{-3}$. We adopt two

pathway databases, namely KEGG 2021 Human and WikiPathways 2024 Human to convert each cell (i.e., 5029 gene) into a set of hundreds of tokens named biological pathways (i.e., 216 tokens derived from WikiPathways and 142 from KEGG).

## A.3 IMPLEMENTATION DETAILS

The experimentation were run on a cluster. We used 900 GB RAM, and Nvidia RTX 8000 GPU. All the techniques for single-cell disease classification were written in Python. The operating system and the different libraries used are: Python 3.8, Pytorch 1.11.0, Captum, Scanpy, GSEApy, Linux.

