# OpenReview forum: "Towards By-Design Interpretable Transformers via Modular Interpretability-Guided Alignment"
_ICLR.cc/2026/Conference — Submitted to ICLR 2026_

### Official Review · Reviewer_LPVr · 2025-10-31

**Soundness:** 2
**Presentation:** 2
**Contribution:** 2
**Rating:** 2
**Confidence:** 3

**Summary:**

This paper proposes a by-design interpretable Transformer framework called Interpretability-Guided Alignment (IGA) to enhance the explainability of biological prediction models. Instead of relying on post-hoc XAI methods, the model integrates a conditional interpretable layer and block-wise interpretability modules that align internal representations with established biological knowledge from databases like KEGG and WikiPathways. Applied to two Alzheimer’s disease single-cell RNA sequencing datasets (Seattle and ROSMAP), the method achieves strong classification accuracy while producing biologically meaningful, context-aware explanations, demonstrating the efficacy of the proposed framework.

**Strengths:**

1. Instead of treating interpretability as an afterthought (as post-hoc XAI does), this work embeds it into the model's architecture and training objective.
2. A comprehensive evaluation is performed with multiple baselines,  including standard attention mechanisms, popular post-hoc XAI methods (Integrated Gradients, DeepLIFT, etc.), and architectural alternatives (Mixture-of-Experts).
3. The fact that their method maintains comparable predictive accuracy while outperforming all baselines on the biological interpretability metric provides compelling evidence for their claims.

**Weaknesses:**

1. The entire framework is critically dependent on the existence of a high-quality "ground truth token annotations vector" which critically limits its generalizability to other domains with no such a priori knowledge.
   - It makes the paper title untenable, which makes it sound like a general framework/procedure for making Transformer architecture more interpretable.
2. The model is trained to align with this external knowledge. What if that knowledge is incomplete, biased, or incorrect? The model would be faithfully interpretable, but wrong, and its alignment could mask novel, data-driven discoveries that contradict the (flawed) ground truth.
3. While the paper does a good job comparing against external baselines but fails to conduct any ablation studies to better assess the significance of their proposed module and loss.
4. Lack of qualitative examples to support interpretability claims beyond numerical scores.
5. While the proposed module and loss are formally defined the underlying intuition is not sufficiently covered.

**Questions:**

1. In Equation 6, you define Output_3 = Weights_i as the Aggregation(W_global + C_i * W_context). Given that W_global and W_context are defined as 2D matrices of size (|P|+1)x(|P|+1), and the resulting Output_3 must be a 1D vector of size (|P|+1) to be compared against the 1D GT_tokens vector in your MSE loss function, what specific Aggregation function is used here? How are these 2D matrices reduced to the required 1D vector?
2. In Section 3.3, you state, "the interpretability-guided alignment step is performed using a randomly sampled subset comprising 50% of the total training observations". What is the rationale for this 50% subsampling?
3. The model-level sub-module in Equation 3 uses a simple summation to aggregate the outputs from the N Transformer blocks (Σ E_block_j). Your analysis in Section 4.4 suggests that different blocks can have specialized roles (e.g., "block 1 pinpoints... than block 2"). Does this simple summation not risk obscuring these distinct, specialized contributions? Did you experiment with alternative aggregation methods?

---

### Official Review · Reviewer_3j7n · 2025-10-31

**Soundness:** 2
**Presentation:** 2
**Contribution:** 1
**Rating:** 2
**Confidence:** 3

**Summary:**

This paper introduces the Interpretability-Guided Alignment Module, a neural layer designed to align the internal activations of pre-trained transformers with task-specific knowledge. The Interpretability-Guided Alignment Module utilizes feedback from additional domain annotations to align the intermediate and aggregated representations of a pre-trained transformer with labels that encode important concepts for the task at hand (e.g., biological pathways). Specifically, this work explores and evaluates the proposed module for transformers trained on biological tasks in the field of neuroscience. This paper evaluates the proposed methodology on two real-world Alzheimer's disease datasets, demonstrating that the resulting models can achieve competitive predictive performance while remaining interpretable and capturing known biological features.

**Strengths:**

I believe these are the paper’s main strengths:

1. **[Significance, Major]** Although the proposed methodology is evaluated only on neuroscience tasks, as the paper rightly mentions, its applicability can easily extend to tasks beyond those in neuroscience. Hence, I believe this work can be of importance for those hoping to build highly predictive transformer-based models for tasks where interpretability is essential. Because of this, and the fact that this work lies within the overlap of interpretability, biology, and neural architecture design, I believe this work has a good chance of being of interest to a large proportion of the ICLR community.
2. **[Quality, Major]** Even though this paper evaluates the proposed methodology only on two datasets, the evaluation is generally detailed and carefully thought through on those two datasets. Moreover, the interpretability results are promising, showing increased interpretability with respect to some reasonable metrics without significant sacrifices in predictive performance.
2. **[Originality, Minor]** The exact mechanism proposed for encouraging alignment between the pre-trained transformer and domain-specific knowledge is, to the best of my knowledge, novel. Nevertheless, as explained below, I have some concerns regarding the novelty of this work in comparison to the extensive research on concept-based interpretability.
4. **[Clarity, Minor]** The paper's writing is generally easy to follow, although, as I mentioned below, some of the technical components could've been better motivated and described.

**Weaknesses:**

In contrast, I believe the following are some of this work’s limitations:

1. **[Originality, Critical]** Although I do believe the specific approach taken by this work was novel, I also think the general intention and the general approach taken are very similar to those of previous methods (particularly those in the concept-based interpretability literature, e.g., [1-5]). In particular, I find the lack of comparison against these methods, and more importantly, the absence of any acknowledgement of these approaches, surprising given the overlap in motivation, approaches, and aims.
2. **[Quality, Critical]** The lack of comparison against any other inherently interpretable models makes the interpretability results not entirely fair (as the models being compared against were not designed to be interpretable by construction and do not take the same "level" of supervision during training). Therefore, as further described below, I believe this work would significantly benefit from comparing itself against other interpretable-by-design concept-based approaches (e.g., [1-5]) or other popular tabular interpretable models (e.g., TabNet [6], TabTransformer [7], or even something like TabCBM [8], which combines both tabular models and concept-based interpretability even for genomics tasks).
3. **[Clarity, Major]** Some of the key contributions and components of the proposed methodology are not very clearly explained, making it hard to fully understand the details of how and, more importantly, why specific components in the proposed module are the way they are. In particular, I believe this paper would significantly benefit from better motivation (carefully explaining why every component was designed the way it was created in the grand scheme of things before describing it) as well as better notation (both mathematical notation and naming conventions).
4. **[Clarity and Significance, Major]** It is very unclear how specific hyperparameters were selected, and no code was provided as part of the submission. Both of these omissions may compromise the reproducibility of the results in this work.
5. **[Significance, Minor]** The focus on only two very niche domain-specific datasets means that it is unclear whether this architecture would work, as claimed in this work, on other domains (as no evidence is provided for that claim).

### References

1. Koh et al. "Concept bottleneck models." ICML (2020).
2. Rigotti et al. "Attention-based interpretability with concept transformers." ICLR (2021).
3. Espinosa Zarlenga et al. "Concept embedding models: Beyond the accuracy-explainability trade-off." NeurIPS (2022).
4. Yuksekgonul et al. "Post-hoc concept bottleneck models." ICLR (2023).
5. Oikarinen et al. "Label-free concept bottleneck models." ICLR (2023).
6. Arik et al. "TabNet: Attentive interpretable tabular learning." AAAI *(*2021).
7. Huang et al. "TabTransformer: Tabular data modeling using contextual embeddings." *arXiv* (2020).
8. Espinosa Zarlenga et al. "TabCBM: Concept-based interpretable neural networks for tabular data." TMLR (2023).

**Questions:**

Balancing the strengths and weaknesses mentioned above, I am leaning towards rejecting this work due to its lack of valuation and situation within the extensive literature of interpretable-by-design neural models. However, I am more than happy to be convinced that some or all of my conclusions are incorrect and to revise my recommendation based on a discussion with the authors. For this, the following questions could help clarify/question some of my concerns:

1. **[Critical]** Could you please elaborate on why the concept-based approaches and tabular-specific methods discussed above have not been discussed in the paper or included as baselines? How is the proposed methodology different from existing concept-based approaches (e.g., CBMs [1]) and, more specifically, tabular-specific methods such as TabNet [7], TabTransformer [7], and TabCBM [8]?
2. **[Critical]** Is it really fair to compare the interpretability of the proposed methodology only against methods that were not designed with interpretability in mind (therefore depending on post-hoc approaches to obtain any helpful explanations)? Similarly, is it fair to compare methods that receive extra training-time feedback (e.g., external knowledge) against those that do not? If not, then how would the interpretability scores of your proposed method look against methods that were designed with interpretability in mind such as TabNet and/or TabCBM? In particular, note that concept-based approaches such as TabCBM can also intake domain-specific knowledge, which could then be used to ensure alignment of its latent space with known task “concepts”.
3. **[Major]** How were the different hyperparameters selected, and how sensitive are the results to these hyperparameters? For example, why is 50% fo the total training set used for the “interpretability-guided step” and how important is that? The appendix just mentions that “several experiments were conducted to select the optimal hyper-parameters” but this is not enough to ensure fair a evaluation and reproduce the results. This is against good scientific practices.
4. **[Minor]** Why, against standard good practices, was no code included as part of the evaluation or discussed anywhere in the paper?
5. **[Minor]** Is there no weight hyperparameter to control the strength of each term in the loss term (e.g., in a similar fashion to CBMs [1])?
6. **[Minor]** In lines 133-135, it is stated that masked gene expressions are zeroed out. Given that zero has a key meaning in gene expressions (i.e., lack of expression), shouldn’t masking set the expression to its empirical mean rather than a zero value?
7. **[Minor]** Do you have any evidence that the proposed architecture could “work” in non-biological datasets?

### Other Suggestions and Typos

1. **[Clarity, Major]** When explaining the overall method, I would strongly suggest better motivating each of the module’s components, perhaps with a running example that clarifies how everything fits together. For example, a big weakness of the way the paper is written is that it is never very clear **how** the external feedback is provided (e.g., what format it takes, how it looks in practice, etc.). Having an example that clearly illustrates this can make a significant difference in how easily it is to follow this work.
2. **[Notation, Major]** I would strongly recommend using `\text{…}` when writing multi-letter words in LaTeX (e.g., $\text{Activations}_i$ vs ${Activations}_i$ and $\text{Model}^\text{Trained}$ vs ${Model}^{Trained}$). Otherwise, the mathematical notation looks cumbersome and harder to read. Moreover, for the sake of clarity, I would suggest using more descriptive names for variables and elements in the architecture that clearly indicate their purpose (e.g., $\text{Output}_1$ conveys nothing about what that output represents).
3. **[Figure Clarity, Major]** I would suggest reworking Figure 1, as right now it is not the easiest figure to follow and does not provide much clarity on the importance and role of each component of the proposed module. To improve it, consider using an example in the figure itself to showcase, hypothetically, what each component of the module would do.
4. **[Results Clarity, Major]** In my opinion, Figure 2 would be much easier to see as a table rather than a figure (the actual visual aspect only adds to the difficulty when comparing results rather than clarity).
5. **[Potential Typo, Minor]** The sentence fragment “… by aligning their internal representations, weights with established …” is hard to parse and seems to be grammatically off.
6. **[Reading Flow, Nit]** Given that the introduction starts by discussing LLMs, but then the text moves directly into transformers, the reading flow is a bit convoluted (if someone doesn’t know what a transformer is w.r.t. an LLM, this text does nothing to help you understand it). Therefore, it may be worth starting the discussion with transformers and then explaining how they serve as the backbone for LLMs.
7. **[Potential Typo, Nit]** Sentence 103 (”In the literature, scSimilarity (Heimberg et al., 2025) model for cell analysis employs an MLP backbone …”) seems off and does not naturally parse.
8. **[Potential Typo, Nit]** In 196-197, “Each element of the vector represent a token …” should probably be “Each element of the vector represents a token …”
9. **[Potential Typo, Nit]** At the end of the caption of Figure 5, there is a dangling colon.
10. **[Reading Flow, Nit]** The sentence in lines 439-441 is very hard to read/parse thoroughly.

---

### Official Review · Reviewer_BErJ · 2025-11-01

**Soundness:** 3
**Presentation:** 2
**Contribution:** 3
**Rating:** 4
**Confidence:** 3

**Summary:**

The paper introduces a Modular Interpretability-Guided Alignment framework that makes Transformer models interpretable by design rather than relying on post-hoc explainability techniques. The method enhances pre-trained Transformers by aligning internal representations and weights with external biological knowledge, particularly in the context of Alzheimer’s disease. Experiments on two real-world  Alzheimer’s disease datasets,  show that the model matches baselines accuracy while tripling biological interpretability scores.

**Strengths:**

* The authors presents an approach that integrates domain-specific knowledge into model predictions, enabling meaningful and interpretable explanations, and evaluates its performance on real-world datasets. This work demonstrates how interpretability methods can be adapted to real-world problems to enhance model reliability.

* The proposed model achieves performance comparable to baseline methods on real-world datasets while demonstrating substantially higher interpretability than traditional post-hoc explanation techniques.

* The presentation is clear also for readers without a background in biology.

**Weaknesses:**

*  The presentation of the architecture is difficult to follow. Specifically, it would help to clarify the Shared Conditional Interpretable Layer, its purpose, and why the conditional mechanism (C_i) is necessary. Additionally, Figure 1 is hard to follow

* Some architectural details are missing. The paper does not specify: (1) the total number of parameters in the model, (2) the exact number of transformer blocks used. There is also a contradiction regarding the base model: Section 3.1 states "we fine-tune a Transformer backbone", while Section 4.4 claims "we defined our own transformer architecture... trained from scratch".

**Questions:**

I would be happy to reconsider my scores after receiving these clarifications:
1. Could you please clarify the role of the Ci in the Shared Conditional Interpretable Layer ?
2. Something is not Clear to me about the output 3, is output 3 is input-dependent? In addition why rank(Weights) is interpretable?
3.  Please provide more details about  (1) number of Transformer blocks, (2) total of parameter count, (3) Whether the backbone is fine-tuned or trained from scratch.


Minor typo: you wrote loss_cassification instead of loss_classification

---

### Official Review · Reviewer_48YN · 2025-11-01

**Soundness:** 1
**Presentation:** 1
**Contribution:** 1
**Rating:** 2
**Confidence:** 4

**Summary:**

The authors design a novel, biologically inspired interpretable conditional layer designed to improve interpretability of internal transformer-based representations. They evaluate their architecture on two cell datasets, Seattle and ROSMAP. The authors find that their method produces more biologically valid and interpretable representations while retaining predictive performance.

Overall, the paper is very difficult to follow and lacking important details. Most notably, Section 3 is overly complex and written in math mode mainly with text, where formulaic descriptions of layers would suffice. Some of the named components are unnecessary: the “aggregation block” is a sum over all inputs. The “projection layer” is a 1-d convolution. Others are just not defined properly.
In the shared conditional interpretable layer: CI_{proj_i} (first part of eq 5) — what is the Shared_layer function? Furthermore, ”the second term is activated only when the context of input tokens is deemed important” — how, and when is the context of the input tokens deemed important? This is not explained in the paper. The output-level sub-module consists of a classification head, shared layer output (unnecessarily given a different name), and a sum over weight matrices, which is unclear how it is used.

Another major concern is that the base Transformer model is not mentioned — is a module trained from scratch? Which model was used? These are questions which need to be answered within the paper.

Finally, I would appreciate more information on the data used in the paper. The gene data is introduced in L132, but the datasets are not described or delineated properly. It would help if prior to the first reference to dataset specifics both the datasets and pathway databases were explained.

It is my opinion that the paper requires significant revisions in clarity and content to reach a publishable state.

**Strengths:**

- The studied problem of explaining transformer internals is important.

**Weaknesses:**

- The writing would benefit from improvement, especially in the method exposition.
- Some very important methodological details are missing or underdefined, making it hard to assess the importance of the results.

**Questions:**

See review

---

### Meta-Review · Area_Chair_Qnfg · 2025-12-05

**Summary:**

The reviewers highlighted multiple shortcomings with this paper, including a lack of clarity around the architecture proposed and insufficient validation in other domains or comparisons with other baselines.

I was in particular compelled by Reviewer's 3j7n questions about the lack of mention of concept-bottleneck methods or other transformer architectures with built-in interpretability. I would encourage the authors to carefully consider the reviewers' comments.

**Reviewer Concerns:**

There was no response from the authors.

**Reviewer Scores:**

Scores were low for this paper and I do not believe the paper is of sufficient quality based on the reviews.

---

### Decision · Program_Chairs · 2026-01-26

Reject